# Out-of-State Travel for Abortion among Texas Residents following an Executive Order Suspending In-State Services during the Coronavirus Pandemic

**DOI:** 10.3390/ijerph20043679

**Published:** 2023-02-19

**Authors:** Gracia Sierra, Nancy F. Berglas, Lisa G. Hofler, Daniel Grossman, Sarah C. M. Roberts, Kari White

**Affiliations:** 1Population Research Center, University of Texas at Austin, Austin, TX 78705, USA; 2Advancing New Standards in Reproductive Health, Department of Obstetrics, Gynecology and Reproductive Sciences, University of California, Oakland, CA 94612, USA; 3Department of Obstetrics and Gynecology, University of New Mexico, Albuquerque, NM 87131, USA; 4Steve Hicks School of Social Work, University of Texas at Austin, Austin, TX 78705, USA

**Keywords:** abortion, out-of-state travel, abortion restrictions, Texas, United States

## Abstract

During the COVID-19 pandemic, existing and new abortion restrictions constrained people’s access to abortion care. We assessed Texas abortion patients’ out-of-state travel patterns before and during implementation of a state executive order that prohibited most abortions for 30 days in 2020. We received data on Texans who obtained abortions between February and May 2020 at 25 facilities in six nearby states. We estimated weekly trends in the number of out-of-state abortions related to the order using segmented regression models. We compared the distribution of out-of-state abortions by county-level economic deprivation and distance traveled. The number of Texas out-of-state abortions increased 14% the week after (versus before) the order was implemented (incidence rate ratio [IRR] = 1.14; 95% CI: 0.49, 2.63), and increased weekly while the order remained in effect (IRR = 1.64; 95% CI: 1.23, 2.18). Residents of the most economically disadvantaged counties accounted for 52% and 12% of out-of-state abortions before and during the order, respectively (*p* < 0.001). Before the order, 38% of Texans traveled ≥250 miles one way, whereas during the order 81% traveled ≥250 miles (*p* < 0.001). Texans’ long-distance travel for out-of-state abortion care and the socioeconomic composition of those less likely to travel reflect potential burdens imposed by future abortion bans.

## 1. Introduction

The onset of the COVID-19 pandemic disrupted the provision of sexual and reproductive healthcare services across the globe. Healthcare providers and advocates worked to modify models of care and change policies to ensure people could obtain comprehensive services, including abortion care, while minimizing risk of exposure [1,2]. Countries with fewer abortion restrictions were more likely to introduce changes that facilitated access to abortion care than countries where abortion was severely restricted; stay-at-home orders and travel bans further compromised pregnant people’s access to abortion in these settings [1,2].

In the United States (US), similar divergent policy responses related to abortion access occurred at the state level. In 12 states, policy makers leveraged the pandemic to enact policies that limited access to abortion care [3]. Texas was one of these states, and at the time had one of the most restrictive abortion policy environments in the US. After the onset of the pandemic, Texas Governor Greg Abbott issued an executive order on 22 March 2020, stating that all surgeries and procedures not medically necessary to correct a serious medical condition or to preserve the life of a patient should be postponed for a period of 30 days [4].The next day, the state’s attorney general declared that the order prohibited most abortions, contrary to guidance from professional medical associations that abortion care should not be delayed [5,6]. During the same period, other abortion-specific executive orders and adaptive responses to COVID-19 itself in Texas’ neighboring states of Arkansas, Louisiana and Oklahoma disrupted access to abortion care [3,7].

Previous studies found that hundreds of Texas residents traveled out of state during the executive order period to obtain abortion care [8,9]. However, beyond documenting overall increases in out-of-state travel, studies assessing the impact of the executive order—as well as studies about prior restrictions—did not examine people’s travel patterns, geographic travel flows and shifts over time or whether people obtained care at the nearest out-of-state location [9,10,11,12]. A detailed analysis of these patterns would illustrate the burdens of out-of-state travel and the disproportionate effects that abortion restrictions have on people needing care in different communities. Additionally, now that the US Supreme Court has ended federal abortion protections, and states—including Texas—have banned abortion [13], these analyses can provide insights about where Texans might obtain care, as well as how short-lived abortion bans affect the timing of out-of-state travel.

In this study, we expand on these prior analyses of Texans’ out-of-state travel for abortion, using individual-level data on Texas residents who obtained abortion care in Arkansas, Colorado, Kansas, Louisiana, New Mexico, and Oklahoma following implementation of the 2020 state executive order. In addition to assessing temporal changes in people’s travel in response to the order, we examined whether Texans obtained care at the nearest out-of-state health center and how travel varied according to geographic area of residence and indicators of county-level economic disadvantage. We hypothesized that increases in out-of-state travel would be delayed following the order and that many Texans would not obtain care at the nearest out-of-state health center. We also hypothesized that travel would be more common among those who lived in more (versus less) advantaged areas.

## 2. Materials and Methods

### 2.1. Data

This study draws on three data sources: de-identified individual-level data on Texas residents who traveled out of state for care, mystery client calls to office-based abortion facilities, and community measures of economic well-being from the Distressed Community Index [14]. We received individual-level data on Texas residents who obtained abortions from organizations operating 25 of the 38 open facilities in nearby states between 1 February 2020, and 31 May 2020. Four other facilities in these states did not provide individual-level data and reported serving between 0 and 22 Texas residents per month during the study period. We were unable to obtain individual or monthly data from nine facilities, most of which were located in Colorado. Health centers provided us with data that they routinely collect as part of electronic health records and/or mandatory reporting to their state’s vital statistics office. Individual-level data included date and type of abortion, gestational duration at time of abortion, patient’s reported zip code, and city or county of residence. Because we did not have zip code information for 25% of our observations, we used county of residence for all of our analyses. We matched county of residence for 99.4% of the observations for which we had zip code or city. Information on patient age and race/ethnicity was not consistently available, so we did not include it in these analyses.

We collected data on whether abortion facilities were open and scheduling appointments using mystery client calls placed between 2 April 2020, and 8 July 2020, to all facilities in Arkansas, Kansas, Louisiana, New Mexico, and Oklahoma. Following a structured protocol, research assistants made weekly calls in April and biweekly calls between May and early June. Callers contacted each facility during regular business hours up to three times over three consecutive days or until successful contact was made, whichever occurred first, and entered information into a standardized form at the end of each call. Callers documented whether a facility was open and scheduling appointments (versus closed or open but not scheduling appointments) and the number of days until the next available appointment (i.e., wait time). If a facility did not answer after three attempts, we considered it to be closed or not scheduling.

To address our hypothesis that travel patterns would vary among those who lived in more (versus less) disadvantaged areas, we used data from the 2020 Distressed Communities Index (DCI) as a proxy measure of the economic well-being of Texas residents. The DCI measures economic well-being and inequality at the zip code and county level [14]. The economic deprivation scores are used to group geographic areas into quintiles: prosperous, comfortable, mid-tier, at risk and distressed. We used county-level distressed community index (Figure 1).

This study was conducted according to the guidelines of the Declaration of Helsinki, and approved by the Institutional Review Boards (IRBs) at the University of Texas at Austin (protocol 2011-11-0025; approved 6/3/2020) and University of California, San Francisco (protocol 14-15184; approved 2/11/2015; and protocol 13-11384; approved 8/2/2013).

### 2.2. Measures

We categorized Texas residents’ abortions into three periods: date of abortion occurring before the executive order was issued (1 February 2020–21 March 2020), during the order (22 March 2020–21 April 2020) and after the order expired (22 April 2020–31 May 2020). We calculated the one-way travel distance from the population-weighted centroid of patients’ county of residence to abortion facilities in Texas and out-of-state facilities in nearby states using the *georoute* module in Stata 15 (StataCorp) [15]. We identified the nearest in-state and nearest out-of-state facility, based on shortest distance traveled, and computed the difference between the nearest facility and facility where people obtained abortion care.

We used mystery client call data to determine whether the nearest out-of-state facility experienced service disruptions in the week prior to a person’s abortion. We considered facilities to have a service disruption if they were temporarily closed/not scheduling or were open with a wait time ≥7 days, because waits ≥7 days might affect a person’s eligibility for different abortion methods. Service disruptions could have been due to adjustments to new COVID-19 protocols, executive orders temporarily banning abortion in certain states (Arkansas, Louisiana, and Oklahoma), or surges in demand for care. For abortions that occurred between two mystery call dates, we used information on service disruptions for the call date that was closest to the abortion date. Between March 22 and April 5, the date we began mystery calls, we assumed facilities had a wait time <7 days, unless they were located in a state that also issued an executive order, in which case we determined abortion availability based on information provided by facility staff.

We categorized abortions by type and gestational duration: medication, procedure ≤ 11 weeks of gestation from last menstrual period, procedure 12–15 weeks, procedure 16–21 weeks, and procedure ≥22 weeks. Because the Texas state limit for abortion in 2020 was 22 weeks, we expected all Texas residents >22 weeks gestation would travel out of state for abortion care, regardless of the executive order.

### 2.3. Analysis

#### 2.3.1. Time Trends in Out-of-State Abortion Care

We conducted all analyses in Stata 15, except where noted. We used chi-squared tests to compare differences according to characteristics of facilities where Texas residents received care, county-level economic deprivation, one-way distance traveled, and abortion type and duration. We calculated the weekly number of Texas-resident out-of-state abortions by gestational duration for the period between 1 February 2020 and 31 May 2020 to assess which changes might be related to the Texas executive order.

Using weekly abortion totals and an interrupted time series (ITS) design, we estimated negative binomial segmented regression models to assess changes in the number of out-of-state abortions after the start of the executive order on 22 March 2020 and following the end of the ban. Interrupted time series analysis is a form of segmented regression that is used to estimate the change in level and trend over time of an outcome between a pre- and post-intervention period [16]. The model coefficients can be interpreted as follows. “Baseline weekly trend prior to implementation of executive order (1 February 2020 through 21 March 2020)” represents the underlying weekly trend in out-of-state abortions before the order was issued. “Implementation of Executive Order (22 March 2020)” is the change in the number of out-of-state abortions in the week immediately following the implementation of the executive order on 22 March 2020 compared to the week immediately prior. “Weekly trend after implementation of the executive order (22 March 2020 through 18 April 2020)” represents the change in the weekly trend in out-of-state abortions from the period before the executive order was issued (1 February 2020 through 21 March 2020) to the time immediately before the ban expired (22 March 2020 through 18 April 2020). “Expiration of the executive order (19 April 2020)” is the change in the number of out-of-state abortions in the week immediately following the expiration of the executive order when health centers resumed full services on 19 April 2020, compared to the previous week. “Weekly trend after the expiration of the executive order (19 April 2020 through 31 May 2020)” represents the difference in weekly trends in out-of-state abortions between the first (22 March 2020 through 18 April 2020) and second (19 April 2020 through 31 May 2020) policy periods. Although the order expired at midnight 21 April 2020, we used 19 April 2020 as the effective end date in order to have equal-length weeks for our analyses.

To account for the time that it might have taken Texans to make travel arrangements, we performed a series of sensitivity analyses. First, we estimated the above-described segmented regression model and used a one-week lag (29 March–4 April 2020) for the effective date of the executive order to allow people to schedule and attend out-of-state appointments. We conducted a second sensitivity analysis that used a one-week lag for the end of the order (26 April 2020 through 2 May 2020) to account for the fact that people might have decided to keep previously scheduled out-of-state appointments or were uncertain when Texas health centers would re-open. In the third sensitivity analysis, we lagged both the start and end dates of the order.

#### 2.3.2. Geographic Flows and Travel Distance

To assess geographic shifts in the origin and destination (i.e., flow) of Texas residents obtaining out-of-state abortion care, we calculated the number and percentage who traveled out of state in each of the three time periods; we grouped Texas’ 254 counties into the state’s eight health service regions to facilitate comparisons between periods and account for the small number of residents traveling from some areas. We used R statistical software to create maps depicting the Texas county-to-facility geographic flows and one-way distance traveled by Texan residents in each period. We also computed the weekly number of Texas residents who traveled, by destination state, to explore changes in patient volume over time that might be related to other states’ executive orders [3].

Finally, we calculated the median distance and interquartile range (IQR) to the out-of-state facility where Texas residents received care and the median distance (IQR) traveled beyond their nearest facility during the executive order period, according to facility proximity/service availability, county-level economic deprivation, and abortion type/gestational duration. For those with gestational durations of ≥22 weeks, we identified the nearest facility as the shortest distance to a facility in Colorado or New Mexico, because those are the only nearby states where abortion was available beyond 22 weeks. We compared differences in median distance traveled for these groups using Wilcoxon rank-sum and Kruskal-Wallis tests.

## 3. Results

### 3.1. Time Trends in Out-of-State Abortions

The number of Texas residents who obtained abortions in nearby states more than doubled between the periods before the executive order was issued and when it was in effect, from 315 to 889 (Table 1). Before the order was issued, 55% of Texas residents obtained care at the nearest out-of-state facility, and only 8% did so during the period the order was in effect (*p* < 0.001). Service disruptions at the nearest out-of-state facility were more common during the executive order period for those who did not obtain care at their nearest location, compared to the period after the order was lifted (69% vs. 37%; *p* < 0.001). A plurality (29%) of Texas residents went to the nearest out-of-state facility after the executive order expired; however, this proportion was significantly lower than during the period before the order was issued (55%, *p* < 0.001).

Texans who resided in the two most economically disadvantaged county quintiles accounted for half (n = 162; 52%) of those who traveled out of state before the order went into effect. The number and percentage of Texans traveling from these areas decreased during the executive order period (n = 102; 12%), and the majority (68%) of Texas residents who traveled out of state during the executive order resided in the most advantaged counties (i.e., those in the top two quintiles). The percentage of Texas residents who traveled ≥250 miles one way to obtain out-of-state abortion care more than doubled between the period before the executive order was issued and when it was in effect (38% to 80%; *p* < 0.001), and then decreased after the order expired, with 61% traveling ≥250 miles one way (*p* < 0.001). Significantly more Texas residents obtained medication abortion (51%) during the executive order than in the periods before (23%) or after (38%; *p* < 0.001). The number of Texans who obtained care ≥22 weeks of gestation did not significantly change over the three time periods.

Figure 2 displays the weekly volume of Texas residents traveling out of state. In the segmented regression models examining weekly variation in the volume of Texas residents who traveled out of state for abortion care, the increase in the number of out-of-state abortions was not statistically significant the week the executive order was issued (incidence rate ratio [IRR] = 1.14; 95% CI: 0.49, 2.63; Table 2, Model 1). However, the weekly trend in the number of Texas residents who obtained out-of-state abortion care was significantly higher during the remainder of the executive order period, relative to the trend observed before it went into effect (IRR = 1.64; 95% CI: 1.23, 2.18). After the executive order expired, the number of out-of-state abortions among Texas residents decreased by 45% (IRR = 0.55; 95% CI: 0.32, 0.95) and continued to decline in the following weeks (IRR = 0.44; 95% CI: 0.33, 0.59). In sensitivity analyses in which the implementation of the executive order was lagged, the number of Texans who obtained out-of-state abortions increased significantly compared to the previous week before the order was issued (Table 2, Models 2–4). In models that lagged the effective end date of the order, both the change immediately following the expiration of the order and subsequent weekly trend remained statistically significant.

### 3.2. Geographic and Travel Distance Patterns in Out-of-State Abortions

Before the executive order was issued, the majority (70%) of Texans who traveled out of state for abortion care lived in East Texas and the Texas Panhandle, regions of the state without an abortion facility (Figure 3, Panel A; Table 3). Over half (55%) traveled to Louisiana and one-third (33%) went to New Mexico. Texans who traveled to New Mexico (n = 105) resided across the state, and many (n = 44; 42%) traveled ≥500 miles one way to obtain care.

During the executive order, the majority (83%) of Texas residents who obtained out-of-state abortions resided in or around counties where in-state facilities had suspended services (Figure 3, Panel B). The primary destinations for Texas residents were New Mexico (31%) and Kansas (25%). Increased volume in these states corresponded with a period during which no Texans obtained abortion care in Oklahoma (Figure 4). Overall, 131 (15%) Texas residents traveled to Colorado and 91 (10%) to Louisiana, most of whom received care during the second half of the executive order period. After the order expired, Texas residents most often traveled to New Mexico (40%), Louisiana (30%) and Oklahoma (13%; Figure 3, Panel C).

During the executive order, the median one-way distance traveled was 189 miles (IQR: 98 to 240 miles) for Texas residents who obtained care at the nearest out-of-state facility, 155 miles further than the median travel distance before the executive order (Table 4). The median one-way travel distance was 468 miles (IQR: 335 to 737 miles) for those whose nearest out-of-state facility had service disruptions. Texans living in the most economically prosperous county quintiles traveled further for care than those who resided in the most distressed counties (median 403 miles versus 289 miles, *p* < 0.001). Texans who obtained care between 16 and 21 weeks of gestation traveled a median of 627 miles one way (IQR: 366 to 811); those who had a gestational duration of ≤11 weeks traveled 384 miles (IQR: 279 to 694). People who obtained abortions ≥22 weeks of gestation (beyond the Texas gestational limit) traveled a median of 689 miles one way (IQR: 649 to 855 miles), similar to the distance traveled before and after the executive order.

## 4. Discussion

Following Texas’ March 2020 executive order temporarily suspending in-state abortion care, the number of Texas residents who obtained abortions out of state increased significantly. Our study provides a nuanced examination of this increase by describing changes in abortion timing and travel patterns as Texas residents identified out-of-state facilities and executive orders in other states disrupted care [3]. That the effects of the executive order on out-of-state travel were not observable until a week or more following its implementation likely reflects the time it took people to identify another facility and make arrangements to travel long distances after local access to abortion care was disrupted [17,18]. The slower decline in out-of-state travel after the order expired might be related to a backlog at Texas abortion facilities as they tried to accommodate patients whose appointments had been canceled and uncertainty about whether the order would be extended, as well as Texans traveling for previously made appointments at out-of-state facilities. Our findings showing a large increase in out-of-state travel after the implementation of abortion restrictions is consistent with other studies [8,11,17]. It also demonstrates that many, but not all, people will overcome significant barriers to obtain care, even following bans on abortion.

By assessing geographic travel patterns, we also observed that the majority of Texans who traveled out of state lived in counties with greater economic advantages; these areas also correspond to large population centers in Texas. Notably, however, the absolute number of people traveling out-of-state for abortion care from the most economically depressed counties decreased during the executive order, suggesting residents of these areas might have had fewer resources to travel long distances during the early COVID-19 pandemic. Moreover, few residents of South Texas counties traveled out of state during the study period, which might be related to the high levels of economic disadvantage and difficulties crossing interior border checkpoints located 100 miles from the US-Mexico border. Although perhaps not be unexpected, these findings indicate that people from border communities—and other areas of economic disadvantage—might not be able to obtain out-of-state abortion care without additional resources to support travel when abortions are banned. It is also possible that people who were unable to travel during this period delayed obtaining care until after the order was lifted [8], ordered medication abortion pills online [19] or obtained medications to end their pregnancy from Mexico.

For those Texans who successfully obtained out-of-state care, these findings illustrate the substantial travel burden they experienced. Although residents in some areas of Texas traveled long distances even before the order was issued, travel distances were far greater after the order was implemented. The majority of Texans did not obtain care at their nearest out-of-state facility, further increasing travel distance. The latter finding is likely related, in part, to service disruptions at facilities in nearby states related to other executive orders [3,7] and modifications in clinical practice to mitigate the real public health risk of exposure to COVID-19 prior to the availability of vaccines and treatment [20]. With few facilities providing abortion care in Arkansas, Louisiana, and Oklahoma, people were forced to travel even further because no other facilities were available. The fact that many people did not obtain care at the nearest facility might also reflect other factors that people consider when seeking abortion, such as referrals from trusted sources, differences in abortion restrictions between states, and variation between facilities in the types of abortion offered, gestational limits, and cost [21,22]. For example, people might have decided to travel further to obtain care in Colorado or New Mexico, states with fewer abortion restrictions and lower median costs for first-trimester abortion care [23], which many people have to pay out of pocket because insurance coverage for abortion is limited.

These geographic patterns and travel burdens offer a partial view of how abortion access might shift now that Texas and surrounding states are allowed to prohibit abortion following the US Supreme Court decision in June 2022 that overturned constitutional protections for abortion established under *Roe v. Wade*. Texans who have previously been able to obtain care in Arkansas, Louisiana and Oklahoma can no longer do so because these states have banned abortion [11,17,24]. As we observed, when abortion becomes inaccessible in those states, Texans will likely travel to Colorado, Kansas, New Mexico, and states further away [25]. Kansas and New Mexico have few abortion facilities that will likely be overwhelmed by residents of other states also seeking care, which could lead to long wait times for appointments [26]. Although not observed in these data given the short time period in which the order was in effect, long waiting times, combined with delays arranging out-of-state travel, might result in people being unable to obtain an abortion [27] or obtaining abortion care later in pregnancy [26]. Abortions later in pregnancy are very safe, but they can require additional visits, can increase cost and length of stay, and are associated with a higher risk of complications compared to abortion care earlier in pregnancy [28]. Additionally, these results highlight the disruptive effects of very restrictive abortion laws that are implemented for even a short period of time, which has become more common as state-level bans are litigated.

This partial view of changes in abortion access following more restrictive policies in the US also has global relevance. The rise of conservative far-right governments in European countries such as Poland, Hungary, and Italy has undermined people’s ability to obtain abortions in their country, which could prompt people to travel elsewhere for care, as seen here [29,30]. Studies from Europe have demonstrated the burdens of cross-country travel for abortion and, like this analysis, have found that people from advantaged backgrounds are more likely to overcome travel barriers, [31,32] leading to unequal access to care [33]. Such disparities in access also persist in countries, such as Mexico, that have liberalized federal abortion policies, but where many states still have not changed their penal codes that criminalize abortion [34]. Moreover, the rise of very restrictive abortion policies in Texas and other US states might further energize anti-abortion groups’ attempts to roll back other countries’ abortion liberalization efforts, using funding they receive from U.S.-based organizations or through the arm of US international policy with the election of a more conservative administration [35,36,37].

### 4.1. Limitations

We did not obtain individual-level data from all facilities in nearby states; travel patterns and distance estimates might be different for those Texans who sought care in other locations or other states. Additionally, we did not have zip-code level information for all records and relied on county of residence to estimate distance and economic disadvantage. As a result, travel distances are likely greater and economic disadvantage would be more variable than what we measured. We used one-week lagged values for facility service disruption at the nearest out-of-state facility as a proxy for the disruption status at the time patients made their appointment. Thus, we are unable to determine the exact service disruption status at the nearest out-of-state facility at the moment each person decided where to obtain care and cannot precisely assess how service disruptions affect care seeking. Relatedly, our data do not allow us to assess the motivation for why Texas residents obtained care where they did.

### 4.2. Conclusions

Despite these limitations, this study provides a detailed assessment of out-of-state travel for Texans obtaining abortion care during this period. In addition to distance traveled, our analysis of county-to-facility flows and service disruptions demonstrates the uneven travel patterns and abortion availability following Texas’ executive order during the early onset of the COVID-19 pandemic and how access to abortion care changed in this period. These patterns also offer insight about individual travel burdens and the strain on out-of-state facilities after Texas and many neighboring states prohibited abortion following the June 2022 Supreme Court decision overturning *Roe v. Wade*.

## Figures and Tables

**Figure 1 ijerph-20-03679-f001:**
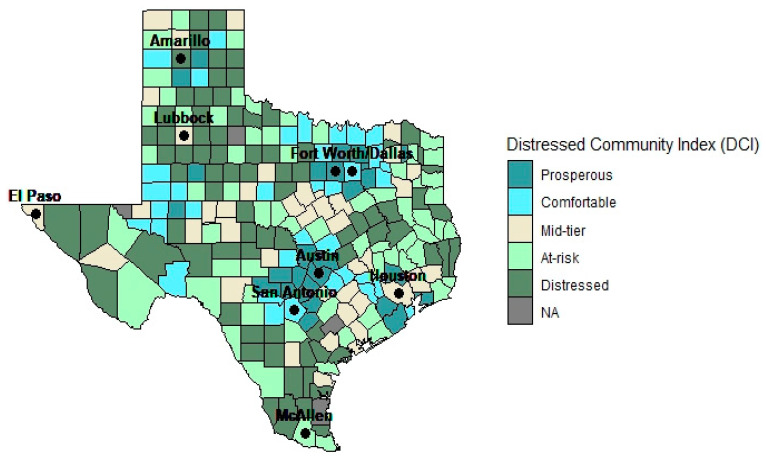
Distribution of Texas counties, by distressed community index. The Distressed Community Index measures economic well-being and inequality. It combines seven distinct and complementary socioeconomic indicators, measured at the county level, to create a single score. The score is used to sort each county into quintiles that depict how the economic well-being in the county compares to other areas: prosperous, comfortable, mid-tier, at risk and distressed.

**Figure 2 ijerph-20-03679-f002:**
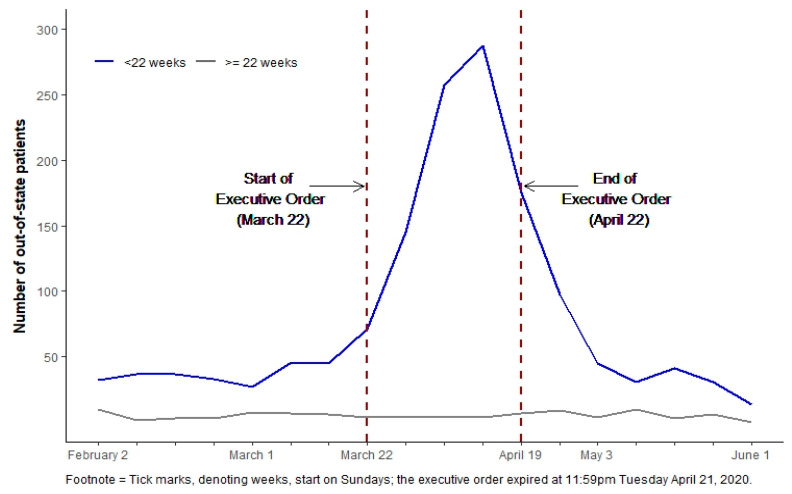
Weekly volume of Texas residents obtaining out-of-state abortion care, by gestational duration.

**Figure 3 ijerph-20-03679-f003:**
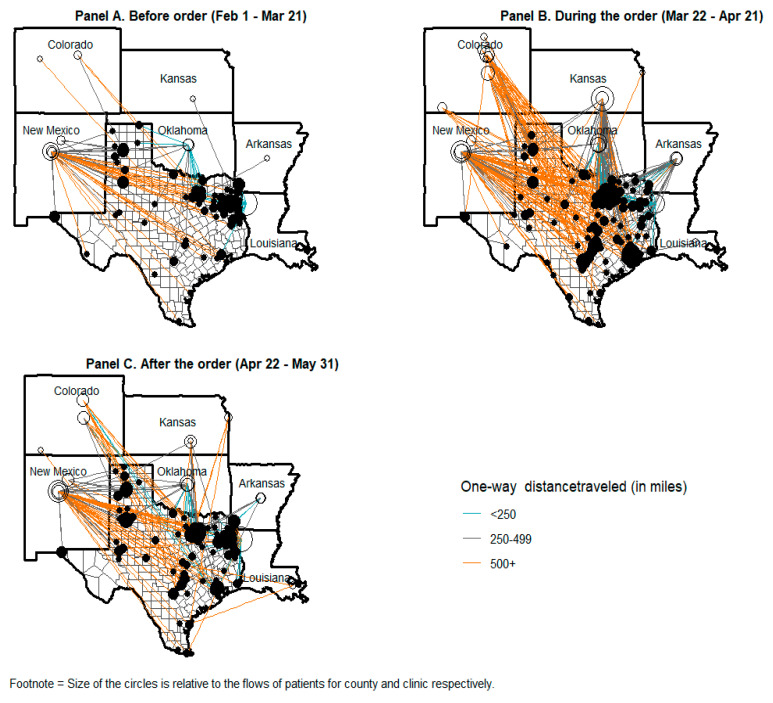
Flow of Texas residents obtaining out-of-state abortion care, by executive order period 1 February–31 May 2020.

**Figure 4 ijerph-20-03679-f004:**
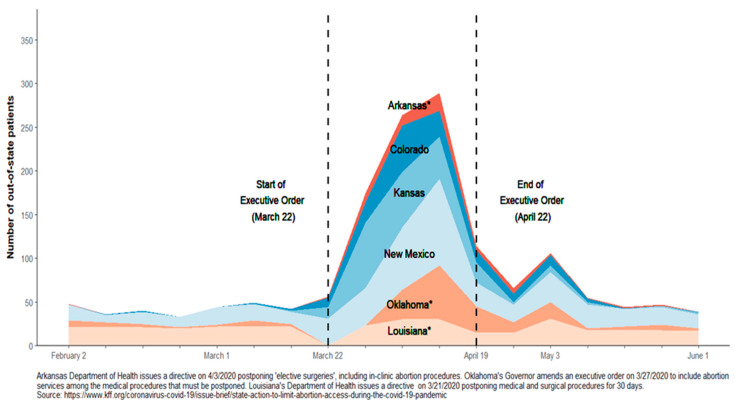
Weekly volume of Texas residents who obtained out-of-state abortion care, by state.

**Table 1 ijerph-20-03679-t001:** Characteristics of where Texas residents obtained out-of-state abortion care and type of abortion obtained, by executive order period 1 February–31 May 2020.

	Before Order was Issued (1 February–21 March)	While Order was in Effect (22 March–21 April)	After Order Expired (22 April–31 May)
**All Texas residents, n**	315	889	341
**Facility where received care, n (%) *****			
Went to nearest out-of-state facility	174 (55.2)	75 (8.4)	100 (29.3)
Did not go to nearest facility	137 (43.5)		
Service disruptions at nearest	--	614 (69.1)	127 (37.2)
No service disruptions at nearest	--	186 (20.9)	112 (32.9)
Missing	4(1.3)	14(1.6)	2 (0.6)
**County-level economic deprivation, n (%) *****			
Distressed	37 (11.8)	30 (3.4)	35 (10.3)
At-risk	125 (39.7)	72 (8.1)	69 (20.2)
Mid-tier	74 (23.5)	178 (20.0)	85 (24.9)
Comfortable	35 (11.1)	263 (29.6)	64 (18.8)
Prosperous	40 (12.6)	333 (37.4)	86 (25.2)
Missing	4(1.3)	13(1.5)	2(0.6)
**One-way distance traveled, miles, n (%)**			
<250	192 (61.0)	170 (19.1)	130 (38.1)
250–499	71 (22.5)	318 (35.8)	111 (32.6)
≥500	48 (15.2)	388 (43.6)	98 (28.7)
Missing	4(1.3)	13(1.5)	2(0.6)
**Abortion type & gestational duration, n (%)**			
Medication	72 (22.9)	455 (51.2)	125 (36.7)
Procedure, ≤11 wks	134 (42.5)	214 (24.1)	104 (30.5)
Procedure, 12–15 wks	48 (15.2)	106 (11.9)	49 (14.4)
Procedure, 16–21 wks	22 (7.0)	89 (10.0)	30 (8.8)
Procedure, ≥22 wks	39 (12.4)	23 (2.6)	32 (9.3)
Missing	0(0.0)	2(0.2)	1(0.3)

*** Chi-squared *p*-value < 0.001, comparing differences between categories and time periods.

**Table 2 ijerph-20-03679-t002:** Estimated change in number of out-of-state abortions among Texas residents, 1 February–31 May 2020.

	Model 1 ^a^		Model 2 ^b^		Model 3 ^c^		Model 4 ^d^	
	Order Implementation & Expiration Not Lagged		Order Implementation Lagged		Order Expiration Lagged		Order Implementation & Expiration Lagged	
	IRR	(95% CI)	IRR	(95% CI)	IRR	(95% CI)	IRR	(95% CI)
Baseline weekly trend prior to implementation of executive order ^e^	1.01	(0.96, 1.06)	1.04	(0.99, 1.10)	1.01	(0.97, 1.06)	1.03	(0.98, 1.09)
Implementation of executive order ^f^	1.14	(0.49, 2.63)	3.64	(2.13, 6.21)	1.91	(0.57, 6.37)	4.41	(2.39, 8.14)
Weekly trend after implementation of the executive order ^g^	1.64	(1.23, 2.18)	1.10	(0.89, 1.34)	1.29	(0.89, 1.88)	0.99	(0.78, 1.26)
Expiration of the executive order ^h^	0.55	(0.32, 0.95)	0.74	(0.46, 1.17)	0.27	(0.08, 0.90)	0.47	(0.26, 0.86)
Weekly trend after the expiration of the executive order ^i^	0.44	(0.33, 0.59)	0.63	(0.49, 0.80)	0.61	(0.41, 0.90)	0.77	(0.60, 1.00)

IRR: Incident rate ratios from negative binomial regression models that adjust for time trends. ^a^ Model 1 assesses changes in the number of out-of-state abortions after the implementation of the executive order on 22 March 2020 and changes following the expiration of the executive order on 19 April 2020 when health centers resumed full services. ^b^ Model 2 assesses changes in the number of out-of-state abortions after a one-week lag in the implementation of the executive order (29 March 2020) to account for the time it might have taken Texans to make travel arrangements once the executive order went into effect. We used 19 April 2020 as the date when the executive order expired and health centers resumed full services. ^c^ Model 3 assesses changes in the number of out-of-state abortions after the implementation of the executive order on 22 March 2020 and a one-week lag (26 April 2020) for the expiration of the order because people might have kept previously scheduled out-of-state appointment or were uncertain whether Texas facilities would resume services. ^d^ Model 4 assesses changes in the number of out-of-state abortions using one-week lags for both the implementation and expiration of the executive order. ^e^ “Baseline weekly trend prior to implementation of executive order” represents the underlying weekly trend in out-of-state abortions prior to the implementation of the executive order. The weekly trend spans from 1 February–21 March 2020 in Models 1 and 3 and 1 February–28 March 2020 in Models 2 and 4. ^f^ “Implementation of executive order” is the change in the number of out-of-state abortions in the week immediately following the implementation of executive order, compared to the week immediately prior to the executive order. The implementation date is 22 March 2020 in Models 1 and 3 and 29 March 2020 in Models 2 and 4. ^g^ “Weekly trend after implementation of the executive order” represents the difference in weekly trend in out-of-sate abortions between the pre and post executive order periods. The weekly trend after implementation of the executive order spans from 22 March–18 April 2020 in Models 1 and 3 and 28 March–18 April in Models 3 and 4. ^h^ “Expiration of the executive order” is the change in the number of out-of-state abortions in the week immediately following the expiration of the order compared to the week immediately prior to the expiration of the order. The date of expiration of the executive order is 19 April 2020 in Models 1 and 2 and 26 April 2020 in Models 3 and 4. ^i^ “Weekly trend after the expiration of the executive order” represents the change in the weekly trend of out-of-state abortions after the order expired compared to the period that the order was in effect. The weekly trend after the order expired spans from 19 April 2020–31 May 2020 in Models 1 and 2 and 26 April–31 May 2020 in Models 3 and 4.

**Table 3 ijerph-20-03679-t003:** Distribution of Texas residents’ region of origin and state where received care, executive order period 1 February–31 May 2020.

	Before Order Was Issued (1 February–21 March)	While Order was in Effect (22 March–21 April)	After Order Expired (22 April–31 May )
	N = 315	N = 889	N = 341
**Health Service Region**			
Panhandle (Lubbock)	54 (17.1)	50 (5.6)	71 (20.8)
North Central (Dallas/Ft. Worth)	54 (17.1)	398 (44.8)	103 (30.2)
East (Tyler)	165 (52.4)	54 (6.1)	81 (23.7)
Southeast (Houston)	13 (4.1)	170 (19.1)	28 (8.3)
Central (Austin)	3 (1.0)	109 (12.2)	20 (5.8)
South Central (San Antonio)	4 (1.3)	63 (7.1)	10 (2.9)
West (El Paso)	16 (5.1)	24 (2.7)	19 (5.6)
South (Harlingen/McAllen)	2 (0.6)	8 (0.9)	7 (2.1)
Missing	4(1.3)	13(1.5)	2(0.6)
**Destination State**			
Arkansas	1 (0.3)	48 (5.4)	11 (3.2)
Colorado	5 (1.6)	131 (14.7)	28 (8.2)
Kansas	1 (0.3)	222 (25.0)	18 (5.3)
Louisiana	172 (54.6)	91 (10.2)	101 (29.6)
New Mexico	105 (33.3)	271 (30.5)	137 (40.2)
Oklahoma	31 (9.8)	126 (14.2)	46 (13.5)

**Table 4 ijerph-20-03679-t004:** One-way distance traveled to out-of-state abortion facilities during Texas’ executive order 22 March–21 April 2020, by select characteristics.

	Median Distance (IQR) to Facility where Received Care	Median Distance (IQR) beyond Nearest Facility
**Facility where received care *****		
Went to nearest out-of-state facility	189.2 (97.8, 239.8)	155.2 (0, 239.2)
Did not go to nearest facility		
Service disruptions at nearest	468.0 (334.8, 736.6)	442.7 (305.6, 706.1)
No service disruptions at nearest	403.2 (334.0, 695.1)	360.1 (218.5, 678.0)
**County-level economic deprivation *****		
Distressed	289.4 (285.4, 413.4)	46.5 (27.1, 179.9)
At-risk	304.2 (120.4, 673.2)	186.0 (9.2, 622.6)
Mid-tier	450.1 (304.1, 884.0)	444.4 (239.2, 883.4)
Comfortable	381.1 (324.1, 652.9)	360.1 (309.8, 646.7)
Prosperous	402.7 (334.0, 695.1)	381.6 (305.6, 688.9)
**Abortion type & Gestational duration *****		
Any abortion, ≤11 wks	384.2 (279.2, 693.6)	360.1 (204.1, 683.4)
Procedure, 12–15 wks	366.3 (334.0, 650.0)	360.1 (247.6, 631.4)
Procedure, 16–21 wks	627.0 (366.3, 811.0)	617.8 (348.7, 716.4)
Procedure, ≥22 wks	688.6 (648.9, 854.8)	9.9 (9.9, 9.9)

IQR: Interquartile range. *** Kruskal-Wallis test *p*-values < 0.001.

## Data Availability

Data are not publicly available owing to confidentiality protections.

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
