# Peer review of "Out-of-State Travel for Abortion among Texas Residents following an Executive Order Suspending In-State Services during the Coronavirus Pandemic"

_ijerph, 2023, doi:10.3390/ijerph20043679_

Round 1

Reviewer 1 Report

In the United States, state-level abortion restrictions force some pregnant people to travel for abortion care. This study assessed Texas abortion patients’ out-of-state travel patterns before and during the implementation of a state executive order that prohibited most abortions for 30 days in 2020.

The authors used data on Texans who obtained abortions between February and May 2020 at 25 facilities in six nearby states. They found that the number of Texas out-of-state abortions increased 14% the week after the order was implemented (incidence rate ratio IRR=1.14; 95% CI: 0.49 – 2.63) and increased weekly while the order remained in effect (IRR=1.64; 95% CI: 1.23 – 2.18). In addition, residents of the most economically disadvantaged counties accounted for 52%, and 12% of out-of-state abortions before and during the order, respectively (p<.001). The policy implication of the paper indicated that certain groups of the population might suffer the abortion bans more because those living in more economically disadvantaged areas will be less likely to travel long-distance for abortions.

 Specific comments.

The manuscript is interesting. The study is timely and fills a significant policy-related research gap. Therefore, I read it with a strong interest. I have the following comments:

1.     The author Sarah C. M Roberts has superscripts 1 and 3. There was no author affiliation for 3. Please fix that.  

2.     The author’s affiliation for superscript 1 needs details. Please add the departments and school for 1, following the format for 2.

3.     The reviewer’s major concern is how reliable and valid the three data sources are. In particular, there is no reliability or validity regarding the individual-level data on Texas residents who obtained 75 abortions from organizations operating 25 of the 38 open facilities in nearby states between February 76 1, 2020, and May 31, 2020.

4.     The mystery client calls' validity and reliability are also questionable.

5.     It could be very problematic if the authors used those self-reported “fake” zip codes to construct the distressed community index.

6.     Travel distance/patterns models were not even valid if based on faked zip codes.  

7.     The statistical analyses seemed justified and adequate if the data is valid.

The discussion and conclusion section seems adequate. 

Reviewer 2 Report

Dear authors,

I found your research of great interest facing the nowadays reality. Banning abortion rights for any woman is a hush facet of extremist laws. Even though such rulings intervene with the freedom of choice, challenging complications can surface whenever an intervention is postponed. I read with great interest your article, as the Supreme Court decision profoundly changed the context of our specialty as gynecologists. 

I believe your material might benefit from making some light on the cost problem. Is there any difference between states in the amount charged for the abortion procedure that might supplementary explain why some Texans would travel so far to undertake it? Or, perhaps, the overwhelming number of women addressing out-of-state clinics to terminate a pregnancy produced a blockage and the impossibility of booking a procedure. Are there problems with the insurance companies when applying for reimbursements for such operations in some clinics, or is there no difference in this matter?

Reviewer 3 Report

This work assessed the out-of-state travel patterns relative to the implementation time of abortion restriction policy in Texas after the onset of the COVID-19 pandemic. It is an interesting study supported with a clear presentation of statistical analysis. The methods are described in detail while the limitations of the current study are also presented well. I have a few comments as below.

1.

I suggest the introduction section be strengthened with more references of related studies or can describe in more detail what have been done in related work and how the existing work and respective findings have prompted the authors’ interest to carry out the current investigation. Some ideas have actually been discussed in Ln 43-51, but they can be further explained or supplemented with more background details.   

2.

(p.13) I also suggest a conclusion for this work or supplement the discussion section with how the current findings would provide insights for future abortion policy in relation to the potential burdens that would likely occur in out-of-state travel. It is also desirable if the authors can summarize the findings in the current context and link to a wider perspective in concluding section when imposing relevant policy.

3.

Would consider for Figure 1 (p.7), the label of the arrow on the right to be “effective end date used in analysis: April 19, instead of the written one “End of Executive Order (April 22)” as the arrow is pointing to the time frame of April 19.

Reviewer 4 Report

This article explores interstate travel around abortion procurement for Texan residents. The data collected is of importance, the method is sound, the writing is succinct and appropriate in tone. This is an urgently timely article of high significance to readers internationally, who watch on in fascination at the repeals of women's and LGBTQ+ rights in the USA around abortion and other topics as part of an overall conservative turn under the wake of the Trump Administration's executive orders and changes to the Supreme Court composition.

This is eye-opening material for those who understand the broader international and US context politically; however, a few minor changes to this well-reported study should be put into place to ensure the relevance to the international readership of this journal who may be unaware of such background information is enhanced. Readers need the necessary background to this current state of affairs to comprehend the significance of travel data and the greater meaning of Texan, US and international conservative turns for rights recognitions.

Firstly, the opening of this article (line 27 and the first paragraph or so) really does not take advantage of the opportunity to speak to an international audience in their own terms, and in so doing, to draw them in to this information. It only speaks to a US audience, which is a mistake and will turn off readers who may actually have much to gain from this piece if you could only show people why. It would be useful for example to consider citing research on how US policies around human rights have long been seen as 'the standard' internationally and have had great influence on other nations, and yet are now in decline or being rescinded in ways influencing other nations... this shows an argument of the relevance of a piece on Texas abortion access to somebody reading in Poland, Australia or South Africa for example. A good piece on this dynamic is 'Trump, Trans Students and Transnational Progress' [Jones, T. (2018), Sex Education, VoL. 18, no. 4, 479–494] which includes information on the international policy sway of both progressive and conservative US political groups and parties around rights locally and in other countries, and how the Trump Administration sought to create a major shift in LGBTQ+ rights in the US and overseas through executive orders and supreme court appointments. This leads us to how the same can be said for women's reproductive rights; it is important to note that the executive orders around Title IX (as argued in the piece) and the court appointments created a kind of insecurity around rights protections in the US for women, LGBTQ+ people and others that are now playing out at the state levels in the US (and in other countries). This speaks to the relevance and importance of this piece, the 'why read this'.

The piece then does do a good job of going into the overturning Roe vs Wade in terms of Texas (from line 35) the local state by state impacts (around line 60) so otherwise the literature review was good, it just only served local US populations and this is an international journal with an international readership - so it really must make the case this is of interest to us internationally first and 'also' as readers from elsewhere.

The methods and methodology are clear. The ethics information is appropriately cited. There is plenty of well-presented data and appropriate figures. The data is summarised in a way that might be locally meaningful in the findings section.

However, when we start to close the article around the second half of p.12-13, what is currently lacking again is consideration of the international readership and how these data can be made meaningful to them. It might be important for example to consider how much the vulnerabilities that were in Texan law apply to other contexts? This does not need much by way of lines, but just a few points on whether this is a US-specific problem or under threat of becoming an international one. For example, there have been considerable regressions in the rights of women in Afghanistan, Poland, and some conservative turns in several other countries... but part of the issue of the US was that protections for women and LGBTQ+ people were won through courts and executive orders rather than more traditional processes that are harder to undo. So this issue may be in some ways specific to the US experience, but in other ways laws are vulnerable in democracies around the world and need to be considered this way under threat of every election, war times and other conditions. Some comment on the international relevance of the findings should be made in such lights; international travel around abortion could be briefly noted as an issue for some people.

Otherwise, this is a wonderful piece and really, I just feel we need to ensure it is presented in a way that it attracts more potential attention and shares of the international readership. We need to ensure it contributes to wider thinking and captures the momentum around rights regressions in many contexts for marginal groups at this time; and speaks more profoundly to this. The data and the work has been done, think of the diversity of the reader and the breadth of the final message a little more.
